# Brevinin-2GHk, a Peptide Derived from the Skin of *Fejervarya limnocharis*, Inhibits Zika Virus Infection by Disrupting Viral Integrity

**DOI:** 10.3390/v13122382

**Published:** 2021-11-28

**Authors:** Weichen Xiong, Jingyan Li, Yifei Feng, Jinwei Chai, Jiena Wu, Yunrui Hu, Maolin Tian, Wancheng Lu, Xueqing Xu, Min Zou

**Affiliations:** 1Guangdong Provincial Key Laboratory of New Drug Screening, School of Pharmaceutical Sciences, Southern Medical University, Guangzhou 510515, China; xiongweichen@outlook.com (W.X.); ljy713121@163.com (J.L.); fei987801933@163.com (Y.F.); vivian20@i.smu.edu.cn (J.C.); jienawu0315@126.com (J.W.); yunruihu@smu.edu.cn (Y.H.); maolint163@163.com (M.T.); lu1132802261@i.smu.edu.cn (W.L.); 2Guangzhou Key Laboratory of Drug Research for Emerging Virus Prevention and Treatment, School of Pharmaceutical Sciences, Southern Medical University, Guangzhou 510515, China

**Keywords:** antimicrobial peptides, brevinine, Zika virus, viral inactivator, viral integrity, antiviral agents

## Abstract

Several years have passed since the Zika virus (ZIKV) pandemic reoccurred in 2015–2016. However, there is still a lack of proved protective vaccines or effective drugs against ZIKV. The peptide brevinin-2GHk (BR2GK), pertaining to the brevinin-2 family of antimicrobial peptides, has been reported to exhibit only weak antibacterial activity, and its antiviral effects have not been investigated. Thus, we analyzed the effect of BR2GK on ZIKV infection. BR2GK showed significant inhibitory activity in the early and middle stages of ZIKV infection, with negligible cytotoxicity. Furthermore, BR2GK was suggested to bind with ZIKV E protein and disrupt the integrity of the envelope, thus directly inactivating ZIKV. In addition, BR2GK can also penetrate the cell membrane, which may contribute to inhibition of the middle stage of ZIKV infection. BR2GK blocked ZIKV E protein expression with an IC_50_ of 3.408 ± 0.738 μΜ. In summary, BR2GK was found to be a multi-functional candidate and a potential lead compound for further development of anti-ZIKV drugs.

## 1. Introduction

Zika virus (ZIKV) refers to a member of the family *Flaviviridae*. It was initially isolated in 1947 from a rhesus monkey in the Zika forest of Uganda [1]. In 2015, ZIKV reoccurred as an epidemic disease spreading to at least 33 Central and South American nations [2]. Up till December 2019, ZIKV had been detected in 87 nations, territories, and subnational areas [3]. It has been shown that ZIKV can be transmitted perinatally, sexually, or through blood transfusion [4]. The most common clinical presentations of ZIKV infection consist of fever, rash, headache, joint pain, and conjunctivitis [5]. In addition, ZIKV infection has been directly associated with neurological disorders (e.g., Guillain–Barre syndrome and congenital microcephaly). However, no specific anti-ZIKV drugs have been approved by the FDA till now, and most clinical treatment strategies focus on analgesics and tranquilizers to reduce symptoms [6]. The genome of ZIKV consists of a (+)-strand ssRNA molecule of nearly 10.7 kb, encoding a polyprotein precursor that is cleaved and processed by the viral protease NS3 into seven non-structural (NS) and three structural proteins (i.e., capsid (C), pre-membrane (prM), and envelope (E)). On the whole, the NS proteins are involved in viral polyprotein processing, genome replication, and operation of host responses for viral benefit, while the structural proteins form the virus particle [7].

As opposed to antiviral drugs developed based on small molecules and antibodies, the development of peptide drugs has aroused increasing attention for their better safety and higher activity [8,9]. Over the past few years, peptides exhibiting anti-ZIKV activity have also been reported frequently. Cathelicidin-derived peptides GF-17 and BMAP-18 have been reported to exhibit anti-ZIKV effects through direct inactivation of the virus and via the interferon pathway [10]. Furthermore, the representative analog of snake cathelicidin BF-30, ZY13, was reported to inhibit ZIKV infection in vitro and in vivo [11]. Antimicrobial peptides (AMP) from amphibians are an important source of antiviral peptide drugs [12,13,14,15,16], of which brevinins are a family with significant pharmacological activity. In addition to significant antibacterial activity, brevinins have also been reported to have antiviral activity against HSV, HIV, and ZIKV [17,18,19]. In order to further expand the medicinal values of brevinins, we screened some brevinins for their antiviral activity. In the present study, brevinin-2GHk (BR2GK) from the brevinins family was found to have anti-ZIKV activity, and the potential mechanism was investigated.

## 2. Materials and Methods

### 2.1. Materials

Vero, Hela, and Huh7 cell lines were obtained from ATCC and were cultured in complete Dulbecco’s Modified Eagle’s Medium (DMEM, Gibco, Waltham, MA, USA) containing 10% fetal bovine serum (FBS, ExCell Bio, Shanghai, China), 100 units/mL penicillin plus 100 μg/mL streptomycin (Gibco, Waltham, MA, USA). ZIKV strain (Z16006, GenBank: 955589.1) was a kind gift from Professor Changwen Ke, Guangdong Provincial Center for Disease Control and Prevention, and proliferated by Vero cells. EGCG (Bide Pharmatech, Shanghai, China) and ribavirin (Sigma-Aldrich, St. Louis, MO, USA) were dissolved in anhydrous DMSO and then stored at −20 °C until being applied. Neutralizing antibodies against Zika NS1 protein and Zika Envelope protein were purchased from BioFront Technologies (Tallahassee, FL, USA) and Genetex (Irvine, CA, USA), respectively. HRP-conjugated goat anti-mouse IgG as secondary antibody was purchased from Cell Signaling Technology (CST, Danvers, MA, USA).

### 2.2. Peptide Synthesis

The precursor sequence of BR2GK was identified from *Fejervarya limnocharis*. BR2GK and FITC-BR2GK were synthesized by GL Biochem Ltd. (GL Biochem, Shanghai, China). The crude synthetic peptide was purified with an Inertsil ODS-SP (C-18) reverse-phase HPLC column to exhibit a purity higher than 95%. The high-purity peptide was pooled, lyophilized, and further confirmed by MALDI-TOF mass spectrometry (Shimadzu, Kyoto, Japan), and finally dissolved in water.

### 2.3. Cytotoxicity Assay

Target cells were seeded in 96-well plates and incubated overnight at 37 °C in 5% CO_2_. Then, the cells were treated with BR2GK at different concentrations. After being incubated for 3 days, MTT solution (0.5 mg/mL) was added and incubated for 4 h. Subsequently, DMSO was added to dissolve the formazan crystals, and the absorbance at 570 nm was measured with a GeniosPro microplate reader (Tecan, Durham, NC, USA).

### 2.4. Plaque Assay

For the antiviral assay, Vero cells were seeded in 12-well plates at a density of 2 × 10^5^ cells per well and incubated overnight. Then, ZIKV or BR2GK was inoculated to the plates and cultivated as indicated. The generation of progeny viruses was evaluated by measuring the reduction in ZIKV-induced plaques. Vero cells at a density of 2 × 10^5^ cells per well were seeded on 12-well plates and exposed to different dilutions of the supernatant from the antiviral assays at 37 °C for 1 h. Next, the viral inoculum was removed, and DMEM containing 2% FBS and 1.2% methyl cellulose was introduced to the cells. After 4 days, the cells were fixed with 4% polyoxymethylene and stained with crystal violet. 

### 2.5. Quantitative Real-Time PCR (qRT-PCR)

Total RNA was extracted with a viral RNA extraction kit (Qiagen, Valencia, CA, USA) according to the manufacturer’s instructions. The cDNA was synthesized and amplified with SuperScript-III (Takara, Shiga, Japan). Then, the ZIKV RNA copies were quantified, and GAPDH acted as an endogenous control to normalize the differences in the total RNA amount in the respective sample. qRT-PCR was performed in triplicate, and the relevant analysis was performed with the 2^−ΔΔCT^ threshold cycle method. The primer sequences are shown in Table 1 [20].

### 2.6. Western Blot Analysis

Vero cells were seeded in 6-well plates at a density of 4 × 10^5^ cells per well and incubated overnight. Then, the cells were infected with ZIKV (100 TCID_50_) for 1 h. The cells were washed and treated with BR2GK at different concentrations. After incubation for 72 h, the cells were collected, and total proteins were extracted with RIPA lysis buffer. Equal amounts of cell lysates were analyzed by 10% SDS-PAGE and then transferred to polyvinylidene fluoride membranes (Millipore, Burlington, MA, USA). The membranes were blocked with 5% non-fat milk and subsequently stained overnight with primary antibody at 4 °C. The following day, the membranes were probed with the secondary antibody for 1 h at room temperature. The protein bands were detected by chemiluminescence using ECL reagent and photographed using a FluorChem R imager (ProteinSimple, San Jose, CA, USA).

### 2.7. Time-of-Addition Assay

Vero cells were seeded in 12-well plates at a density of 2 × 10^5^ cells per well and incubated overnight. Then, the cells were infected with ZIKV (100 TCID_50_). After being incubated for 1 h, 20 μM of BR2GK was added at different times. For the −0.5 timepoint, BR2GK was added to the cell culture 0.5 h post ZIKV infection. The total protein was extracted after 24 hpi, and the relative protein level was quantified by Western blot.

### 2.8. Fluorescence Microscopy

Vero cells were seeded in 48-well plates at a density of 3 × 10^3^ cells per well and incubated overnight. Then, the cells were infected with ZIKV (100 TCID_50_) for 1 h and incubated with BR2GK under different concentrations. After 72 h, the cells were fixed with 4% paraformaldehyde for 20 min, then incubated with anhydrous methanol at 4 °C for 5 min and blocked with 3% BSA for 1 h. After incubation with anti-NS1 antibody at 4 °C overnight, and then with secondary antibody, the nuclear stain DAPI was applied. The fluorescence was observed on a fluorescence microscope (Nikon, Tokyo, Japan).

To determine whether BR2GK traverses the cell membrane, Vero cells were seeded in 24-well plates at a density of 6 × 10^3^ cells per well. The following day, the cells were incubated with FITC-BR2GK at 37 °C for 24 h. Next, the cells were fixed with 4% paraformaldehyde and permeabilized with 0.2% Triton X-100. Lastly, the cells were incubated with DAPI for 10 min to visualize the nuclei and were observed on the fluorescence microscope.

### 2.9. Cell-Based ZIKV Immunodetection Assay

An immunodetection assay of ZIKV-infected cells was performed based on the earlier description [21]. Vero cells were seeded in 96-well plates at a density of 8 × 10^3^ cells per well and incubated overnight. Then, the cells were infected with ZIKV (100 TCID_50_) for 1 h and incubated with BR2GK under different concentrations. After incubation for 72 h, the cells were fixed with 4% paraformaldehyde for 20 min, followed by permeabilization with ice-cold methanol at 4 °C for 5 min. Next, the cells were blocked with 2% non-fat milk at 37 °C for 1 h and then incubated with anti-ZIKV E antibody (1:1000 dilution) at 37 °C for 1 h followed by incubation with goat anti-rabbit HRP conjugated antibody (1:2000 dilution) at 37 °C for 1 h. The cells were further washed quintic with washing buffer, incubated with TMB for 5 min, and then 1 mM sulfuric acid was added to stop the reaction. The optical density (OD) was measured at 450 nm using a GeniosPro microplate reader (Tecan, Durham, NC, USA).

### 2.10. Virucidal Assay

BR2GK and ZIKV (4 × 10^5^ PFU/mL) were incubated at 37 °C for different times. Subsequently, the mixture was diluted 100 times with serum-free DMEM to reach a concentration of peptide far below the IC_50_ and used to infect Vero cells for 1 h. The antiviral level was detected by plaque assay.

### 2.11. RNase Digestion Assay

BR2GK at different concentrations was incubated with ZIKV (1000 PFU/mL) at 37 °C for 2 h. Then, the mixture was digested by micrococcal nuclease (New England BioLabs, Ipswich, MA, USA) at 37 °C for 1 h to release the RNA from the virus [22]. Afterward, RNA was extracted, and qRT-PCR was performed using the above method.

### 2.12. Molecular Docking

The docking model of BR2GK and ZIKV envelope protein (PBD code: 5IRE) was generated using the ZDOCK 3.0.2 [23,24]. The model of BR2GK was established by SWISS-MODEL [25].

### 2.13. Statistical Analysis

All data are expressed as mean ± standard deviation (SD). A one-way analysis of variance (ANOVA) was used for data analysis. *p* < 0.05 was considered statistically significant.

## 3. Results

### 3.1. BR2GK Inhibits ZIKV Infection In Vitro

First, the cytotoxicity of BR2GK was assessed. In the experimental concentration range (0–20 µM), BR2GK exhibited no significant toxicity to Vero, Hela, and Huh7 cells (Figure 1A). Thus, the interference of cell viability on subsequent experiments was eliminated. Next, four methods were adopted to treat the cells, and the possible anti-ZIKV effects of BR2GK were evaluated by the plaque assay. To be specific, BR2GK was added to Vero cells after preincubation with 100 TCID_50_ ZIKV for 30 min (virus-treatment), 1 h prior to virus infection (cells-treatment), concurrently with virus inoculation (co-treatment), and after virus infection plus washing with PBS to remove external viruses (post-treatment), as illustrated in Figure 1B. According to the results, BR2GK, at a concentration of 20 μM, significantly reduced the number of plaques induced by ZIKV in the virus-treatment, co-treatment, and post-treatment methods (Figure 1C). The determination of ZIKV genomic RNA, E protein, and NS1 protein showed consistent results (Figure 1D–F). The above results suggest that BR2GK has a significant inhibitory effect on ZIKV infection. Since the inhibitory effect was only found when BR2GK was in contact with the virus, its mechanism is more likely to target the virus rather than the cell.

### 3.2. BR2GK Inhibits the Early and Middle Stages of ZIKV Infection

To further evaluate the effect of BR2GK against ZIKV, a time-of-addition assay was performed to determine the role of BR2GK on the life cycle of ZIKV. As shown in Figure 2A,B, the strongest inhibition effect of BR2GK was observed at −0.5 and 0 h, suggesting that it targeted an early stage in ZIKV biogenesis. In addition, BR2GK also significantly inhibited the level of ZIKV E protein at 4 and 8 h, indicating that the peptide may also affect the middle stage of ZIKV infection. On that basis, different concentrations of BR2GK were added 1 h after ZIKV infection to observe its effects. Direct observation of the cytopathic effect (CPE) under the microscope showed that BR2GK could protect cells from ZIKV-induced CPE in a dose-dependent manner, suggesting a potent anti-ZIKV activity of BR2GK (Figure 2C). As indicated by the qRT-PCR and Western blot, the mRNA levels and protein contents of ZIKV E and NS1 protein decreased in a dose-dependent manner after BR2GK treatment (Figure 2D,E). The ZIKV replication inhibitor ribavirin at a concentration of 50 μM, as a positive drug [26], exerted a weaker effect compared with BR2GK at a concentration of 20 μM. Additionally, the immunofluorescence detection of NS1 protein showed a consistent result (Figure 2F). Moreover, a cell-based immunodetection assay was performed to detect the relative content of E protein, and the IC_50_ of BR2GK to inhibit ZIKV infection was determined as 3.408 ± 0.738 μM (Figure 2G). Furthermore, the virus supernatant was collected 72 h after the infection for the plaque assay to detect the production of progeny virus. According to the results, BR2GK reduced ZIKV-induced plaque formation dose-dependently (Figure 2H). As revealed in the mentioned results, BR2GK effectively reduced ZIKV RNA replication and protein synthesis, which may inhibit the early or mid-stage of ZIKV infection.

### 3.3. BR2GK Directly Inactivates ZIKV

Since BR2GK exhibited the strongest activity in the early stage of ZIKV infection, it was investigated whether BR2GK directly inactivated the virus. ZIKV and BR2GK were incubated directly at room temperature for different times and then added to the cells for infection. Plaque formation and cell-based immunodetection assay were exploited to determine the direct effect of BR2GK on ZIKV. Figure 3A presents the result of the plaque formation. BR2GK exhibited strong direct inactivation activity against ZIKV, and it worked immediately after coming into contact with the virus. As indicated by the results of the cell-based immunodetection assay, BR2GK added to the cells simultaneously with the virus inoculation resulted in a 90% reduction in the NS1 protein level. With 5 min co-incubation of BR2GK and ZIKV before inoculation, the inhibition rate of NS1 protein was close to 100%, which was comparative to the positive drug EGCG (Figure 3B). Considering the direct and rapid effect of BR2GK, it is suspected that BR2GK may target the virus membrane and disrupt the integrity of the envelope. In the presence of an intact envelope, externally added micrococcal nuclease cannot degrade ZIKV RNA. In contrast, once the viral envelope was disrupted by different concentrations of BR2GK, varying degrees of RNA reduction occurred (Figure 3C). As a positive control, 1% Triton was added, which dissolved all membranes and proved that the concentration of micrococcal nuclease used was sufficient to degrade ZIKV RNA. In addition, BR2GK was labeled with FITC and incubated with Vero cells. The FITC-BR2GK was observed to penetrate the membrane and enter the cells (Figure 3D). Taken together, BR2GK inactivated ZIKV by disrupting the integrity of the envelope and entered the host cells by penetrating the membrane.

### 3.4. BR2GK Targets ZIKV Envelope Protein

Considering that BR2GK can directly inactivate ZIKV, it is suspected that BR2GK can bind to the envelope protein of ZIKV. Consistently, the results of molecular docking showed that BR2GK could bind to the ZIKV envelope protein, and the highest score of ZDOCK was 1559. This model showed that BR2GK bound tightly to ZIKV E stem domain (Figure 4). Taken together, the above results indicated that BR2GK inactivated ZIKV by binding with ZIKV E stem domain.

### 3.5. BR2GK Inhibits ZIKV Infection in Human Cell Lines

Finally, the inhibitory effect of BR2GK on ZIKV infection of human cell lines was evaluated. BR2GK had no obvious toxicity to human liver cancer Huh7 cells and human cervical cancer Hela cells at 20 μM (Figure 1A). The experimental results of Huh7 cells showed that 1 h after ZIKV infection, 20 μM BR2GK treatment significantly inhibited the mRNA levels and protein content of ZIKV E protein and NS1 protein in infected Huh7 cells and inhibited the formation of ZIKV-induced plaques (Figure 5A–C). The test on Hela cells showed a similar effect (Figure 5D–F). 

The above results showed that BR2GK can inactivate ZIKV in the environment of human cell lines and/or penetrate into human cells to inhibit the middle stages of ZIKV infection.

## 4. Discussion

The skin of amphibians is capable of secreting abundant peptides, in which AMPs have been studied most extensively [27]. Besides antibacterial activity, AMPs have been reported to exert many other pharmacological effects, such as anti-oxidation, anti-tumor, anti-septicemic, and anti-inflammatory activities [28,29,30,31,32]. It has been suggested that several AMPs protecting amphibians against pathogens in their habitat exhibit effective antiviral activities [13,14,33,34]. BR2GK pertains to the brevinin-2 family of AMPs. As revealed from existing studies, BR2GK has significantly weak antimicrobial activity, while its truncated peptides exhibit enhanced activity [35]. Nevertheless, the antiviral activity of BR2GK has not been explored as yet. 

The life cycle of ZIKV comprises adsorption and entry into the host cell, then, replication and proliferation, and lastly, release of the progeny virus [36,37,38]. All the stages may be potential targets of antiviral drugs. Our study showed that BR2GK inhibited ZIKV infection in Vero cells and two human cell lines. Somewhat unexpectedly, it seems that BR2GK can exert a strong effect as long as it starts to make contact with virus particles, and it has a potential inhibitory effect on both the early and middle stages of ZIKV infection, which suggests that it may have several targets. Multi-target antiviral agents are not rare in previous studies. EGCG blocked ZIKV entry through binding to E protein [39], as well as targeting ZIKV protease and helicase [40]. Curcumin not only prevented E protein from binding to the cell surface but also inhibited ZIKV replication [41].

AMPs exert their activity primarily by rupturing membranes and lysing bacterial cells [28,42,43]. It is reasonable to assume that its function in the early stage of ZIKV infection is to directly inactivate the virus. This hypothesis was subsequently confirmed. The infectivity of ZIKV was significantly inhibited with BR2GK treatment. Furthermore, the genomic RNA of ZIKV was released and digested by micrococcal nuclease, indicating that BR2GK disrupted the integrity of the envelope. However, it is necessary to investigate which domain of the ZIKV envelope is targeted by BR2GK. This domain is speculated to have some conserved motifs and can be recognized by BR2GK, and it contributes to understanding why amphibians, rich in AMPs, have not been reported to be infected by ZIKV [17]. Molecular docking showed that BR2GK can bind to the E stem domain of the ZIKV envelope protein, which may indicate a direction for further research.

Since BR2GK also functions in the middle stage of ZIKV infection and can penetrate the cell membrane into host cells, we speculate that it may also inhibit the replication stage of ZIKV. However, ITC analysis showed that BR2GK did not effectively bind to RdRp and MTase proteins (data not shown), indicating that BR2GK may have other unknown targets that need to be further explored.

In brief, this study reported that BR2GK, which was derived from *F. limnocharis*, directly inactivated ZIKV by disrupting the integrity of the envelope and may also penetrate the host cell membrane to inhibit the middle stage of ZIKV infection. These results may be beneficial for further development of peptide-derived antivirals against ZIKV infection.

## Figures and Tables

**Figure 1 viruses-13-02382-f001:**
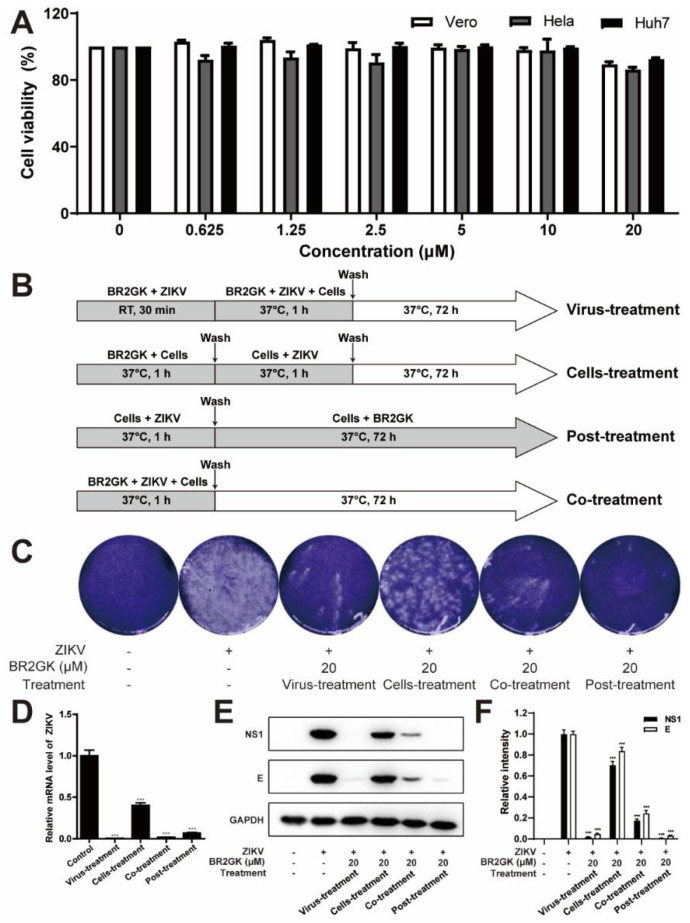
BR2GK inhibited ZIKV infection in vitro. (**A**) The cytotoxicity of BR2GK to Vero, Hela, and Huh7 cells within 72 h was analyzed by MTT assay. (**B**) A schematic representation of BR2GK treatment and ZIKV infection assays using Vero cells (**C**–**E**). Virus-treatment, cells-treatment, co-treatment, and post-treatment were used to treat Vero cells. (**C**) The extracellular virus content was quantified by plaque assay, (**D**) the intracellular ZIKV RNA level was detected by qRT-PCR, and (**E**) the NS1 and E protein levels were detected by Western blot. (**F**) Western blot densitometric analysis. Experimental data are expressed as mean ± SD (*n* = 3), *** *p* < 0.001 compared with the control group.

**Figure 2 viruses-13-02382-f002:**
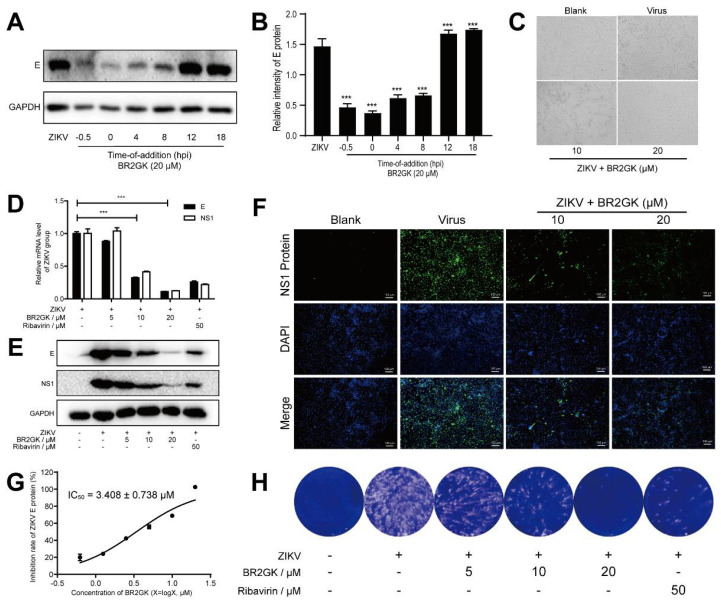
BR2GK inhibited RNA replication and protein synthesis of ZIKV. (**A**) Vero cells were infected with ZIKV for 1 h and treated with 20 μM BR2GK at different time points. The ZIKV E protein was detected by Western blot. (**B**) Western blot densitometric analysis. (**C**) BR2GK reduced virus-induced CPE in Vero cells. Vero cells were infected with ZIKV and treated with BR2GK. After incubation for 72 h, the mRNA and protein expression of E and NS1 were detected by qRT-PCR (**D**) and Western blot (**E**), respectively, with ribavirin as a positive control. (**F**) Expression of NS1 protein was observed by immunofluorescence microscopy. (**G**) Vero cells were infected with ZIKV for 1 h and treated with BR2GK at different concentrations. After incubation for 72 h, the inhibition rate of E protein was detected by cell-based ZIKV immunodetection assay. (**H**) BR2GK reduced the formation of viral plaques. Experimental data are expressed as mean ± SD (*n* = 3), *** *p* < 0.001 compared with the control group.

**Figure 3 viruses-13-02382-f003:**
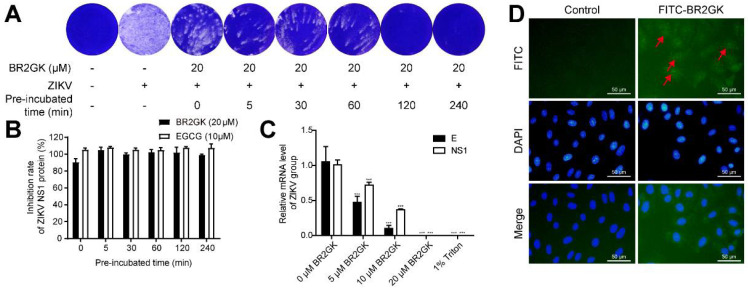
BR2GK directly inactivated ZIKV. ZIKV was pre-incubated with BR2GK for different times before infection; then, the plaque assay (**A**) and cell-based ZIKV immunodetection assay (**B**) were used to evaluate the infection rate of ZIKV. (**C**) BR2GK-treated ZIKV was digested by micrococcal nuclease; then, the degradation of released genomic RNA was detected by qRT-PCR. (**D**) Fluorescence microscopy was used to detect BR2GK (as indicated by arrows) entering Vero cells. Experimental data are expressed as mean ± SD (*n* = 3), *** *p* < 0.001 compared with the control group.

**Figure 4 viruses-13-02382-f004:**
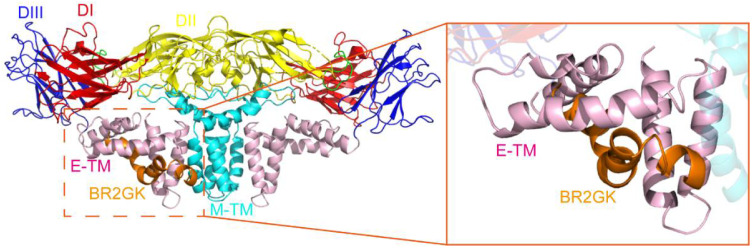
Molecular docking of BR2GK (orange) and ZIKV envelope protein (the domains are colored respectively). A detailed comparison of the binding site of the two structures is depicted in the zoomed view.

**Figure 5 viruses-13-02382-f005:**
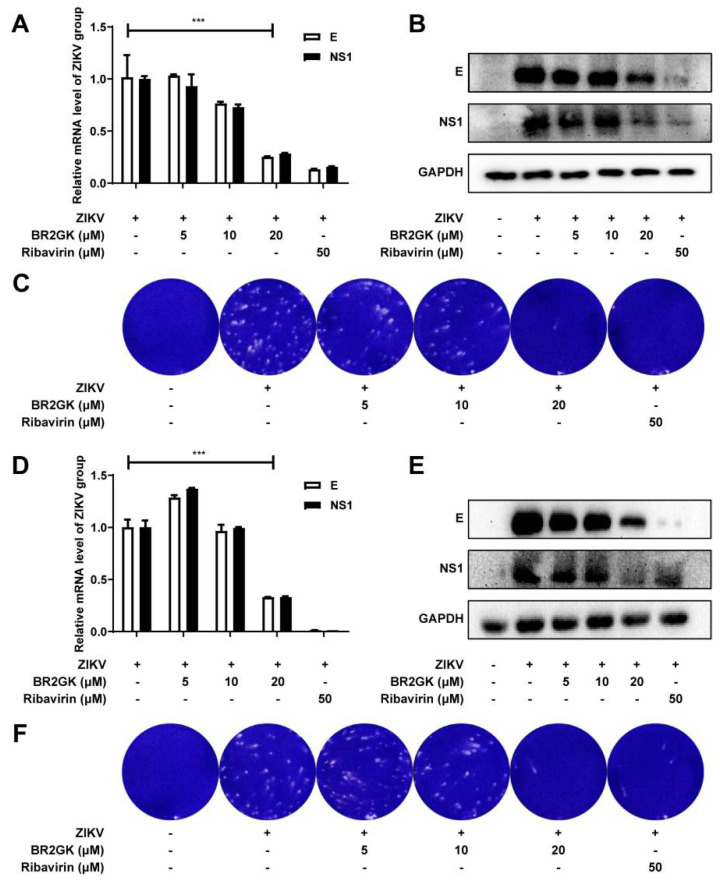
BR2GK inhibited ZIKV infection in human cell lines. (**A**–**C**) BR2GK inhibited ZIKV infection in Huh7 cells. Huh7 cells were infected with ZIKV (100 TCID_50_) and treated with BR2GK. After incubation for 72 h, the cells were collected for gene expression analysis by (**A**) qRT-PCR and protein expression analysis by (**B**) Western blot, and the supernatant was harvested for virus titration by (**C**) plaque assay. (**D**–**F**) BR2GK inhibited ZIKV infection in Hela cells. Hela cells were infected with ZIKV (100 TCID_50_) and treated with BR2GK. After incubation for 72 h, the cells were collected for gene expression analysis by (**D**) qRT-PCR and protein expression analysis by (**E**) Western blot, and the supernatant was harvested for virus titration by (**F**) plaque assay. Experimental data are expressed as mean ± SD (*n* = 3), *** *p* < 0.001.

**Table 1 viruses-13-02382-t001:** Primer sequences for qRT-PCR.

Target	Gene Sequence
ZIKV	F: CVGACATGGCTTCGGACAGY
	R: CCCARCCTCTGTCCACYAAYG
ZIKV E protein	F: GGTGGGACTTGGGTTGATGT
	R: ATGTCACCAGGCTCCCTTTG
ZIKV NS1	F: ACCAGAGAGGGCTACAGGAC
	R: TTAGCCTGGAACGACAGTGG
GAPDH	F: TTGCATCGCCAGCGCATC
	R: TCGCCCCACTTGATTTTGGA

## Data Availability

Not applicable.

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
