# Peer review of "Brevinin-2GHk, a Peptide Derived from the Skin of Fejervarya limnocharis, Inhibits Zika Virus Infection by Disrupting Viral Integrity"

_viruses, 2021, doi:10.3390/v13122382_

Round 1
Reviewer 1 Report
The authors have addressed my comments with satisfaction.
Reviewer 2 Report
Based on the revised version, the author has addressed all of my concerns.
I would like to accept it for publication.
This manuscript is a resubmission of an earlier submission. The following is a list of the peer review reports and author responses from that submission.
Round 1
Reviewer 1 Report
This study showed that the peptide named BR2GK was able to inhibit ZIKA virus replication in a few cell lines. BR2GK showed a powerful inhibitory effect on the ZIKA virus and very little cell toxicity. This peptide could serve as a good candidate for the treatment of Zika virus-infected patients in the future.
The authors properly designed the experiment and presented the results. There are some minor issues the author may need to clarify.
- In Table 1, what is the first primer? Please specify the primer.
- In figure 1D, the results showed that the post-treatment had a stronger inhibition on the viral protein expression than the co-treatment did. I thought the co-treatment should have a stranger inhibition effect because the peptide had more time to interact with the virus. Could you explain this?
- In figure legend 1, there is some information missing. We do not know how long you treat the cells with the peptide.
Reviewer 2 Report
The authors studied the use of BR2GK as a potential drug to inhibit ZIKV infection.
Introduction
The reasoning on why studying this compound is not so strong in my view. Is there more literature that can be included on its effect on other microorganisms? From the abstract I deduce there is literature on fungi? Why did the authors decide to study this compound? Overall, the introduction is rather short.
Materials and Methods.
I am a bit confused with the addition of an infectious dose of 100 TCID50. How does that relate to MOI? MOI 50? For judgement it is even better when you state the absolute number of particles added also because you do state the number of cells. This also holds thru for latter parts (e.g. 2.12, 2.13) were PFU/ml is written.
What is the solvent for the peptide? Water? This as I do not see a solvent control in the listed experiments.
2.10/11 is not an anti-binding/entry assay. This is a typical infectivity experiment.
In many sections (including results), details on when the compound is added and for how long is often lacking
Results.
Figure 1 B
Although it is clear that BR2GK is reducing the number of plaques there is no quantification of the data. This is a representative image but quantitative data should be shown. This remark also holds thru for other figures in the manuscript. Also, based on the results shown for NT in Figure 1 I doubt whether you can still reliable count the number of plaques.
Figure 1D
As in 1B. Quantification is lacking, this should be added as an extra panel.
Figure 2A
RNA levels in the cell supernatant? Or intracellular RNA copies? Have you titrated the vRNA levels at latter time points to investigate the mode of action in more depth? Is this E of NS1?
At what time point, materials were collected for analysis? Also at 72 hpi? This because you will then have multiple rounds of replication and you cannot determine the mechanism. These type of experiments should be performed within 1 round of replication. At 24 hpi re-infection may be abolished and that is why you see a decrease in RNA/protein levels at 72 hpi. Therefore RNA and proteins levels measured at 72 hpi do not provide information on the mode of action and the text should therefore be adapted. It is clear that there is a strong antiviral effect: yet the mechanism is not clear.
205 Language. I do not understand this sentence
Figure 2F IC50 is typically calculated on titers.
Fig 2G quantification is lacking
Section 3.3.
I disagree with the conclusions mentioned by the reviewer. Even though the compounds were only present early in infection; the antiviral effect may be latter if the compound is internalized in cells. In my view the experiments presented here are in fact antiviral infectivity assays and do not provide any information on binding/entry. There are direct assays to evaluate the effect of compounds on cell binding and cell entry. To evaluate the effect on binding you can directly detect the number of bound virus particles by Q-RT-PCR following binding at 4 degrees Celsius (no need to incubate for 72 hr). For cell entry assays, 1 hr incubation at 4 degrees can be followed by a 1 hr incubation at 37 degrees to allow entry. Bound/extracellular virus can be removed by washing/ high salt high pH buffer or proteinase K treatment. The number of internalized particles can be determined via Q-RT-PCR. Thus, if the authors want to claim the potential effect on binding/entry the experiments needs to be repeated.
Additional note after reading 3.4: You cannot distinguish between a virucidal effect or an effect on binding/entry. The above discussed experiments cannot be performed with compounds that are virucidal. With compounds that are virucidal one can only investigate potential additional effects at post-entry conditions and within 1 replication cycle!
Section 3.4
The results clearly show that BR2GK is virucidal. With these results in mind, the earlier cellular experiments are questionable. All of the earlier results can be a direct consequence of the strong virucidal effect. This also holds thru for the post-infection treatment conditions as these experiments were done after multiple rounds of replication (at 24 hpi : newly produced virus maybe inactivated by the compound and therefore cannot infect new cells thereby inducing a decrease in viral RNA/protein at 72 hpi).
Section 3.6. Is BR2GK antiviral in human cell lines OR do you not see infectivity due to the virucidal effect. Given the results presented in Figure 4 it might be due to the virucidal effect therefore one should not state that it is antiviral in human cell lines.
Due to severe concerns regarding the experimental set up/interpretation of the results I do not agree with the conclusions presented in the discussion.